# Morphological Variation in the Striped Field Mouse Across Three Countries

**DOI:** 10.3390/ani15030452

**Published:** 2025-02-06

**Authors:** Linas Balčiauskas, Alexander Csanády, Michal Stanko, Uudo Timm, Laima Balčiauskienė

**Affiliations:** 1Nature Research Centre, Akademijos 2, 08412 Vilnius, Lithuania; laima.balciauskiene@gamtc.lt; 2Department of Ecology, Faculty of Humanities and Natural Sciences, University of Prešov, 17. Novembra 1, SK-08116 Prešov, Slovakia; 3Institute of Parasitology, Slovak Academy of Sciences, Hlinkova 3, SK-04001 Košice, Slovakia; stankom@saske.sk; 4Institute of Zoology, Slovak Academy of Sciences, Dúbravská cesta 9, SK-84506 Bratislava, Slovakia; 5Estonian Environment Agency, Mustamäe Tee 33, 10616 Tallinn, Estonia; uudo.timm@envir.ee

**Keywords:** *Apodemus agrarius*, ecogeographic rules, Slovakia, Lithuania, Estonia

## Abstract

We assessed the variation in morphological traits in the striped field mouse (*Apodemus agrarius*) across Slovakia, Lithuania, and Estonia, aiming to understand how geographic gradients influence body size. The body mass, body length, tail length, hind foot length, and ear length were analyzed from nearly 12,000 specimens collected in several decades. Findings suggest that body size increases toward the north, following Bergmann’s rule, which links larger size to better heat conservation in colder climates. Conversely, appendage sizes (e.g., tail, ear) decrease, aligning with Allen’s rule to minimize heat loss. Longitudinal gradients revealed reductions in body length and appendages toward the east, potentially reflecting climatic and habitat variability. Temporal analysis revealed fluctuating body size trends, possibly due to changing environmental pressures. Therefore, our study underscores the importance of geographic and temporal factors in shaping morphological traits and provides valuable insights into the adaptability of *A. agrarius* to varying climatic conditions.

## 1. Introduction

Body size is a fundamental biological trait that reflects an organism’s adaptation to its environment, with evolutionary trade-offs between body size, resource use, and life-history traits. Larger body size increases vulnerability to scarcity and environmental variability, while smaller size allows persistence in a wider range of conditions, despite typically shorter lifespans and higher mortality risk [1]. It is generally accepted that body size (morphological traits) is constrained by resource use, litter size, predation, etc. [2]. One of the new analyses showed that in some rodents, food abundance and competition had a greater effect on variation in morphological traits than “any reduction in temperature alone, providing little support for ecogeographic rule effects” [3].

Two well-known ecogeographic rules are those of Bergmann (based on body size) and Allen (based on appendages), which state that organisms in colder climates tend to have larger body sizes to conserve heat (Bergmann’s rule) and shorter appendages (ears, tails, limbs) to minimize heat loss (Allen’s rule), whereas those in warmer climates tend to have smaller bodies and longer appendages to promote heat dissipation. In fact, both rules show climatic adaptation in various species to temperature [4], which, in turn, could be translated as latitude. Discussions on datasets and species supporting and not supporting both rules were based on non-universal empirical support [5], with the clines missing in various taxa, e.g., in shrews [6]. As for Allen’s rule, it is supported in rodents, especially for tails, while hind foot length shows no relationship with climatic variables, and ear length shows intermediate patterns related to precipitation. Adherence to Allen’s rule may be influenced by species-specific traits and trade-offs, highlighting the need for further research across taxa to determine the generality of these patterns [7].

Urbanization effects have been shown to modify the influence of temperature on body size in mammals due to better resource availability [8], often resulting in larger individuals. Resource advantages in urban areas underscore the importance of natural history data in understanding anthropogenic morphological change. Therefore, the combined effects of urbanization and changing climatic conditions can summarize their negative impacts [9]. Coupled with energetic constraints in various mammalian species, these changes may have future implications not only for body size, but also for species distributions [10].

Small mammals are a suitable group for broad-scale analyses because they are diverse, have different lifestyles, and are represented in different datasets [8]. Body mass is available for nearly all extant small mammal species [11], and the database is suitable for macroecological analyses and broad perspectives [12]. However, detailed data are not readily available, especially from earlier time periods. As body size is closely related to species biology, the roles of morphological traits should not be underestimated, as they explain the ability to survive in different environments and under climatic changes [13]. The rate of change of morphological traits is no less important [14].

Recent changes in body size have been often linked to global warming, but may also be influenced by resources and other factors [15]. Larger body sizes in some mammals have been driven by habitat transitions associated with climate change, suggesting that regional habitat changes may outweigh climate effects [16]. Rapid and significant changes, sometimes reversing ecogeographic rules, highlight the importance of local studies of small mammals for understanding broader macroecological patterns [17].

As the subject of this study, we choose the striped field mouse (*Apodemus agrarius*), distributed throughout central and eastern Europe, as well as southern Siberia and eastern Asia, with a range disjuncture in the Far East [18]. Genetic studies of many populations of *A. agrarius* from Asia and Europe indicate that it is a relatively new member of the European fauna, with only a single population lineage from Asia having entered Europe. Although several subspecies of *A. agrarius* have been described in Europe based on some morphometric parameters, these populations are genetically much more homogeneous than in Asia [19].

Species expansion in Central and Eastern Europe has been noted in countries such as Ukraine, Czech Republic, Hungary [18], Slovakia [20], and Lithuania [21]. In Estonia, it ranks third in frequency in small mammal monitoring, often dominating grain and vegetable fields [22]. Increasing abundance has a significant impact on other small mammal species [23,24], although the effects remain understudied. For example, in Slovakia, areas newly colonized by *A. agrarius* show lower parasite diversity and host parasitism compared to long-established populations, suggesting that stable parasite–host relationships develop gradually, potentially influencing population health and survival [25]. Thus, parasite–host relationships are gradually established and stabilized, and may influence the health status of *A. agrarius* populations and probably also the duration of survival in a given location when colonizing new areas.

Studies of variation in morphometric traits (body size, appendages, craniometric parameters) have shown that they are useful in the context of habitat-specific factors such as urbanization and pollution. In Poland, it was shown long ago that urban *A. agrarius* have larger cranial dimensions and distinct patterns of size and shape adaptation compared to suburban or rural populations. This fact has been linked to habitat isolation and food availability [26,27]. Thus, increased body weight was considered to be a morphological adaptation [28]. Similarly, changes in body size in relation to industrial pollution were found in Serbia [29].

Geographic differentiation in craniometric and body characters has been studied to a lesser extent. For example, while Bulgarian populations of *A. agrarius* show distinct craniometric patterns that distinguish them as a separate group within Europe, their differences are not so extreme as to imply total isolation [27]. Another study using long-term body condition data of the species in Lithuania [30] did not link morphometric adaptation to climate change or temporal trends.

Analysis of body dimorphism in *A. agrarius* in Slovakia shows that males have larger body dimensions than females, with older individuals having larger body and cranial dimensions [31]. This underscores the importance of considering sex and age in studies of population morphology, and highlights the need for further research into their ecological and evolutionary implications [31].

The study of temporal changes in appendages of *A agrarius* across large latitudinal gradients in China found that absolute appendage lengths (tail, hind foot, and ear) increased significantly over time (1999/2009–2018), indicating that changes were indirectly driven by climate variables. These changes were inconsistent with Allen’s rule, as appendage lengths were more strongly associated with changes in body size than direct temperature effects [32]. However, this study was conducted on *A. agrarius*, which differs in size from the much smaller European subspecies [33].

Most studies of *A. agrarius* focus on habitat-specific or short-term changes, with limited attention paid to large-scale drivers such as climate change or geographic gradients. Morphometric data in Europe are outdated, and comprehensive regional analyses are lacking. This study addresses these gaps by investigating morphological variation in Slovakia, Lithuania, and Estonia, taking into account different environmental conditions and the increasing impact of climate change, which affects size-related variability (e.g., the smallest individuals are in central northern Europe) [33].

This information will be of value in describing changes in the species over time, allowing an assessment of how climate change is affecting the adaptations and survival strategies of the species. Detailed morphological analysis across different geographic regions and long-term time scales will provide insights not only into the evolutionary and ecological processes of *A. agrarius*, but also into the responses of the species to rapidly changing environmental conditions.

## 2. Materials and Methods

### 2.1. Countries and Gradients

Material on *A. agrarius* was collected in three European countries: Slovakia, representing Central Europe, and Lithuania and Estonia, both representing Northern Europe (Figure 1).

Slovakia is situated between 47.73° and 49.62° N, and between 16.83° and 22.57° E. The area of the country is 49,036 km^2^. The altitude ranges from 94 m to 2655 m above sea level. The average temperature in January ranges from −1 to 2 °C, and in July from 18 to 21 °C [34]. Due to the variability of climatic conditions and altitude, the diversity of ecosystems in the country is high [35]. With the prevalence of small settlements, and an urbanized population of 53.5%, Slovakia maintains a favorable balance between natural and human-modified landscapes. Landscapes are dominated by forests and agricultural land (37.8% and 28.6% of the territory, respectively), with grasslands as the third major habitat type (21.1%). Wetlands (0.4%) and inland waters (1.4%) are under-represented [36].

Lithuania is located between 53.90 and 56.45° N, and between 20.93 and 26.85° E. It has a total area of 65,286 km^2^, with lowlands dominating the landscape. In 1991–2020, the average temperature in January ranged from −3.7 to −0.9 °C, and in July from 17.4 to 18.9 °C [37]. In 2024, agricultural land covered 51.6% of the territory, forests 32.9%, water bodies 4.1%, and wetlands 1.5% [38]. In 2024, the country’s population was 2.89 million, of which 68.6% were urban [39].

The territory of Estonia extends from 57.51° to 59.82° N and from 21.76° to 28.21° E, with an area of 45,339 km^2^, of which 9.2% are islands [40]. In 1991–2000, the average temperature in the coldest month, February, ranged from −5.3 to −1.4 °C, and in the warmest month, July, from 17.2 to 18.4 °C [41]. Estonia is a generally flat country with an average height above sea level of about 50 m. In 2022, the main habitats in the country were forests, at 54.1% of the territory; agricultural land covered 28.0%, wetlands 7.8%, and human settlements 7.7% [42]. Current population estimates are about 1.37 million, mainly urban (about 70%) [43].

The three countries are positioned according to the south–north gradient (Figure 1a). Due to the distance between the countries, these parts do not intersect in the latitude expression. In the west–east gradient, the three countries are not completely separated. The eastern part of Slovakia and the western part of Lithuania, as well as the eastern part of Lithuania and the western part of Estonia, intersect in the longitude expression (Figure 1a).

### 2.2. Trapping and Sample Size

In Estonia and Lithuania, *A. agrarius* was trapped using snap traps. In Lithuania, standard snap trap lines were used, with 25 traps of 5 m each in a line. The line was operated for 3 days (rarely shorter), and the traps were checked once a day in the morning or twice a day. Standard traps and the same lines with 25 traps of 5 m each were also used in Estonia, except that the trapping period covered two nights and the traps were usually checked once a day in the morning [22].

In Slovakia, the same snap trap methodology was used until 1996. Since 1996, depending on the research focus of the projects and the type of habitat, ground (pitfall) traps (1996–2002) and live traps (1999–2024) have been used in addition to snap traps. Ground traps were 4 L glass traps buried in the ground up to the mouth of the bottle. In the last twenty years, live trapping has been the dominant method of small mammal research in all habitat types except urban environments [44].

We used retrospective data: the period in Slovakia was 1984–2019, in Lithuania 1981–2024, and in Estonia 1980–1998. Therefore, a direct comparison of *A. agrarius* morphological traits between all three countries was possible for the 1980s and 1990s, and a comparison between Slovakia and Lithuania was additionally possible for the 2000s and 2010s.

The total sample size was 11,928 *A. agrarius* individuals, of which 1101 were from Estonia, 3823 from Lithuania, and 7004 from Slovakia, characterized by differences in demographic and seasonal distributions. Slovakia had a balanced sex ratio and a predominance of subadults, with peak numbers in autumn. The Lithuanian sample was male-dominated, characterized by a high proportion of juveniles and a seasonal peak in autumn. Estonia was also characterized by a male majority, with a predominance of subadults and a seasonal peak in autumn and summer (Table 1).

Standard morphological traits were measured prior to dissection: body mass (Q), body length (L), tail length (C), hind foot length (P), and ear length (A). Mice were weighed to the nearest 0.1 g on spring, weighing, or electronic scales. In pregnant females, the weight of the uterus and embryos was excluded from the body mass measurements. The other traits were measured to the nearest 0.1 mm using mechanical or electronic calipers in a standard manner [45]. Over 80% of the measurements in Lithuania and 85% in Estonia were made by the same person throughout the study period.

Three age groups (juveniles, subadults, and adults) were identified based on sexual organ development, thymus atrophy, and occasionally body mass, which is not a reliable age indicator [46,47]. Juveniles were non-breeding individuals with a well-developed thymus, filiform uterus (females), or abdominal testes (males). Subadults were non-breeding individuals with partially involuted thymus glands and inactive genitalia. Adults showed an atrophied thymus and reproductive activity or post-reproductive traits such as lactation, pregnancy, or developed male reproductive organs (scrotal testes). According to [48], juveniles were excluded from most further analyses.

The habitat distribution of *A. agrarius* specimens showed variations in three countries. Slovakia showed a balanced distribution, with mixed habitats (2315 specimens) and riparian zones (1424) being the most common, followed by forests (1015) and shrublands (888), while meadows were notably rare (4 specimens) and wetlands were not represented at all. Lithuania had a strong dominance of specimens in meadows (2259), followed by a significant presence in commensal habitats (502) and agricultural areas (379), with fewer in forests (234) and wetlands (109). In Estonia, most individuals were found in agricultural habitats (442) and mixed habitats (264), with smaller numbers in forests (138), meadows (147) and other habitat types.

### 2.3. Statistical Analysis

We tested the normality of all traits by decade, age, and sex using the Shapiro–Wilk (W) test, which is particularly effective for small to moderate sample sizes (3–2000) and continuous data, and is better at detecting deviations from normality caused by skewness or kurtosis [49].

Dimorphism (male–female differences in morphological traits) was tested using a series of *t*-tests and grouping individuals by country, decade, and age group. In addition to *t*-tests, we checked whether the male–female differences reached at least a 5% threshold compared to the value of the trait in females. This threshold was calculated as (M − F)/F × 100, where F and M denote trait values in females and males, respectively. Negative values indicate larger trait values in females, and positive values indicate larger trait values in males. In line with R.J. Smith’s criticism [50], we did not use the M/F ratio for our data or for comparisons with other authors’ samples.

To test whether geographic location had a significant effect on body size of *A. agrarius*, we ran a GLM model with Q, L, C, P, and A as dependent factors; decade, animal age, and animal sex as categorical predictors; and latitude and longitude coordinates as continuous predictors. Hotelling’s T^2^ test was used to assess whether variation in latitude and longitude had a simultaneous effect on body size. The significance of each factor was assessed using Fisher’s F-test, F, and p, and effect size was assessed using partial eta-squared (*η*^2^). Furthermore, the relationship between morphological traits and geographic location was presented as correlation and linear regression and visualized graphically.

The normality of distributions was calculated using PAST version 5.0.2 (Museum of Paleontology, Oslo College, Oslo, Norway) [51]. All other calculations were performed using Statistica for Windows, version 6.0 (StatSoft, Inc., Tulsa, OK, USA) [52].

## 3. Results

We found an inconsistent pattern of normality in body size traits in *A. agrarius* that was not related to sex, age, country, or time period (Appendix A). Regardless of age group, males had a higher proportion of normal distributions in traits L and C, while traits Q, P, and A tended to be more non-normal. Similarly, females followed this pattern, with traits L and C showing a greater prevalence of normality, while Q, P, and A were predominantly non-normal. The most notable differences between normal and non-normal distributions occurred in trait Q for males and trait A for females (Table 2).

GLM models show that all analyzed factors were all significant at *p* < 0.0001: latitude (T^2^ = 0.27, *η*^2^ = 0.21), longitude (T^2^ = 0.93, *η*^2^ = 0.07), decade (T^2^ = 0.05, *η*^2^ = 0.21), age (T^2^ = 1.64, *η*^2^ = 0.45), and sex (T^2^ = 0.05, *η*^2^ = 0.05), but of different strength, explaining 63.7% of body mass, 69.7% of body length, 54.7% of tail length, 30.6% of hind foot length, and 20.7% of ear length variation.

Univariate results for each trait, presented in tabular form, indicate that the listed geographic, temporal, and biological factors significantly influenced the morphological traits of *A. agrarius*. Latitude had a strong effect on all traits, especially tail and body length, while the significance of longitude varied. Age was the most influential factor, while sex had no significant effect on body mass and tail length (Table 3).

Based on these results, we further tested for sex dimorphism in adult and subadult individuals by country and decade. We then examined two geographic gradients (south–north and west–east) in the species.

### 3.1. Sex-Based Dimorphism in Apodemus agrarius

The sex-based dimorphism in body mass and length of *A. agrarius* by country and decade is shown in Figure 2. Among adults, females were generally heavier than males, with significant differences in Estonia (10.1% in the 1980s) and Lithuania (ranging from 8.0% to 12.9% between the 1990s and 2020s). In Slovakia, females were 5.7% heavier in the 1990s, but males were 6.3% heavier in the 2010s (Figure 2a). For subadults, no significant differences were observed in Estonia or Lithuania, with weight differences of less than 5%. In Slovakia, subadult males were consistently heavier than females across the decades, with differences ranging from 3.7% to 6.7% (Figure 2b).

Adult *A. agrarius* females were significantly longer than males only in Lithuania, in the 1990s (by 5.4%) and in the 2010s (by 3.8%); all other differences were not significant and of very small magnitude (Figure 2c). In subadult *A. agrarius*, males were longer than females in all countries and periods (Figure 2d). Significantly longer males were found in Estonia in the 1980s and 1990s (by 3.9% and 3.3%), in Lithuania in the 2000s and 2010s (by 1.5% and 2.9%), and in Slovakia in all decades (by 3.2%, 3.2%, 2.6%, and 3.6%, respectively).

Appendage dimorphism in *A. agrarius* was less pronounced than body size (Figure 3). In adult animals, females had longer tails than males, but significant differences were only found in Estonia (1980s, by 5%), Lithuania (2010s, 4.1%), and Slovakia (1980s, 2.7%), while others were not significant, including the 2010s in Slovakia, where males’ tails were longer (Figure 3a). In subadult *A. agrarius*, males had longer tails in three of four decades, with significant differences observed in Estonia (1980s, 2.3%) and Slovakia (2000s, 2.3%; 2010s, 2.0%), all below the 5% threshold (Figure 3b).

Hind foot length was consistently longer in males. In adults, significant differences were found in Lithuania (2000s, 1.7%) and Slovakia (1990s–2010s, 1.5–3.0%) (Figure 3c). Among subadults (Figure 3d), males had longer hind feet in Estonia (1980s, 1.7%), Lithuania (2000s–2010s, 1.2–2.7%), and Slovakia (last three decades, 1.2–3.7%). All differences remained below the 5% threshold.

Ear length in adult (Figure 3e) and subadult (Figure 3f) *A. agrarius* showed no sexual dimorphism, except in the 2010s in Lithuania, where subadult males had larger ears by 2.6%.

### 3.2. South–North Gradient of Body Size in Apodemus agrarius

Morphometric variation along the south–north gradient showed that adult and subadult body mass generally increased northward, except for subadults in the 1980s and adults in the 2000s (Table 4). Body and tail lengths decreased significantly northward, except for non-significant changes in adult body length in the 1990s. Hind foot length showed no trend in the 1980s–1990s, but decreased significantly northward after the 2000s. Ear length increased northward in the 1980s–1990s, but decreased after the 2000s.

To visualize the south–north gradients of various morphological traits, the best examples are presented as scatterplots including a regression line (Figure 4).

### 3.3. West–East Gradient of Body Size in Apodemus agrarius

The variation in body mass along the west–east gradient showed an unstable and generally weak correlation with longitude across decades (Table 5). In contrast, body length, tail length, and hind foot length consistently decreased eastward in adult and subadult animals across all decades. An eastward decrease in ear length was observed in subadults in the 1990s and in both adults and subadults in the 2000s. Figure 5 illustrates west–east gradients of morphological trait variations in *A. agrarius* across decades (1980s–2010s) and age groups (adults and subadults), with regression lines showing trends over time.

### 3.4. Temporal Changes in Body Size of Adult and Subadult Apodemus agrarius

To analyze temporal changes in *A. agrarius*, juveniles were excluded, sexes were pooled, and a GLM model with trapping location coordinates as covariates was used. Temporal trends across three countries could not be fully tested due to limited Estonian data (two decades).

In Slovakia, body mass and length increased from the 1980s to 2000s, then dropped in the 2010s (post hoc, *p* < 0.001). In Lithuania, body mass decreased significantly from the 1980s to 1990s, followed by non-significant increases in later decades. Body length significantly increased in the 2020s, reaching its highest average (Figure 6). In Estonia, body mass changes in the 1990s were non-significant compared to the 1980s, but body length increased significantly (post hoc, *p* < 0.001).

Temporal changes in appendage size of adult and subadult *A. agrarius* in three countries did not follow the same pattern (Figure 7). Tail length in Slovakia fluctuated, and the increase in 2010s was significant compared to the 2000s (post hoc, *p* < 0.001). In Lithuania, the tail length was stable during the 1990s–2020s, as well as in Estonia during the 1980s–1990s.

In Slovakia, hind foot length showed a continuous increase, with significant differences in the 2010s compared to all previous decades. In Lithuania and Estonia, changes in hind foot length were not significant due to high variability. Ear length increased in Slovakia, with a significantly higher average in the 2010s compared to the 1980s and 2000s (Figure 7). In Lithuania, ear length increased after the 1990s, with significant increases in each subsequent decade. Estonia also showed a significant increase in ear length from the 1980s to the 1990s.

## 4. Discussion

Body and appendage size variation in mammals is influenced by adaptation to climatic factors, including climate change [53], as well as resource availability, urbanization, and other factors [9]. Temperature influences body size via latitude and climate change [15], necessitating their joint assessment in the same dataset. While interspecies comparisons could include traits such as pelage color [54], our study focused on a single species, *A. agrarius*, with no data on color differences in the retrospective material.

We chose this species because of documented small mammal response to climate change [55], geographic gradients [32,56], and knowledge from the Baltic region [57]. Our sample is robust and meets key requirements [48]: availability of raw data and sample numbers, use of standard measurements, known measurement accuracy, and assessment of sex and age dimorphism. In addition, the authors measured 80–85% of *A. agrarius* individuals from their respective countries, minimizing bias in the retrospective data.

### 4.1. Overview of Sex-Based Dimorphism and Temporal Trends in Apodemus agrarius

Variations in sexual dimorphism across countries and decades were observed in adults and subadults. Adult females were generally heavier than males, while subadult males often outweighed females, especially in Slovakia. Differences in body length and appendage size (e.g., tail and hind foot) were less pronounced (Figure 2 and Figure 3). In Slovakia, our results are consistent with previous research [44], which reported that adult and subadult males were larger than females in all traits, with significant differences in body length for both groups and in hind foot length for adults. In general, male *A. agrarius* in Slovakia were larger throughout the study period.

Studies on somatic characters of *A. agrarius* in Slovakia have developed since the post-war period, beginning with Kratochvíl and Rosický [58], who presented preliminary data on a limited number of specimens. Subsequent works, such as those by Mošanský [59,60], Dudich and Štollmann [61], and Štollmann et al. [62], expanded the dataset, albeit with varying sample sizes. Stanko and Mošanský [44] provided the largest dataset (*n* = 559) from eastern Slovakia (Východoslovenská rovina plain), providing robust insights into geographic and sexual dimorphism in biometric traits. However, inconsistencies in study design and sample size highlight the need for standardized methods in future research.

In Lithuania, the first very limited data on *A. agrarius* date back to 1934 [63]. Two adult females were larger than the single measured adult male: L = 103–106 vs. 100 mm, C = 82–90 vs. 83 mm, P = 18.4–18.5 vs. 18.2 mm, and A = 12–12.8 vs. 12.2 mm. Large adult individuals of *A. agrarius* were reported in the “Fauna of Lithuania” in the 1980s [64]. Males were smaller than females, mean Q = 27.0 vs. 29.1 g, L = 96.4 vs. 97.6 mm, C = 71.5 vs. 72.4 mm, P = 17.8 vs. 17.9 mm, and A = 12.1 vs. 12.2 mm, respectively. Therefore, in addition to Figure 7, these values indicate a decreasing body size in Lithuanian *A. agrarius* from the 1930s to the 1990s and the subsequent increase.

There are no other published data for comparison, except those from Poland in the 1960s–1970s. Pooled data for all age groups show Q = 20.67 ± 0.17 g, L = 84.46 ± 0.27 mm, and C = 73.68 ± 0.18 mm, with males being significantly larger than females in all these traits [47].

Although the measurements are not directly comparable due to different subspecies, long-term changes in *A. agrarius* traits were confirmed in China: L increased significantly over the period 1995–2015, while Q remained stable over time [65]. All appendages (C, P, and A) increased significantly in length over the period 1999/2009–2018 [32].

### 4.2. Overview of Geographic Gradients of Body Size in Apodemus agrarius

Our results reveal distinct geographic gradients in *A. agrarius* morphological traits. Body mass generally increased northward, while body length, tail length, hind foot length, and ear length decreased, consistent with Bergmann’s and Allen’s rules, reflecting adaptations to minimize heat loss at higher latitudes. However, no data on behavioral or physiological adaptations were available [66]. In contrast, body mass along the west–east gradient was weak and inconsistent, while body length, tail length, and hind foot length consistently decreased eastward, with ear length showing similar trends in specific decades. These patterns suggest that both climatic and ecological factors shaped morphological variation, with thermal constraints driving north–south adaptations.

Publications on geographic gradients in the morphological traits of *A. agrarius* from Europe are not available. In China, *A. agrarius* body length, tail length, and hind foot length decreased significantly with increasing latitude, while body mass remained stable geographically [32,65].

Studies of other *Apodemus* species show variation in body size along geographic gradients. In the herb field mouse (*Apodemus uralensis*), body and skull size showed high variability across the species range. Body mass, zygomatic width, and condylobasal length decreased from south to north (contrary to Bergmann’s rule), increased from west to east, and increased with altitude. Tail, hind foot, and ear lengths were greatest in the southern range, consistent with Allen’s rule [56].

Latitudinal gradients in external and cranial measurements were studied in *Apodemus peninsulae*, *A. draco*, and *A. latronum* across East Asia and Siberia. In *A. peninsulae*, skull and body measurements generally correlated positively with latitude, except in Siberia, Sakhalin, and Hokkaido. *A. draco* showed significant negative correlations in skull and tail measurements at higher latitudes, whereas *A. latronum* consistently showed positive correlations in skull and hind leg lengths across regions [67]. A reanalysis of *A. draco* traits found no support for Bergmann’s or Allen’s rules, with trends in body size and foot/snout ratios contradicting these predictions [68].

### 4.3. Connecting Climate Trends to Morphological Variability in Apodemus agrarius

While Slovakia experienced a range expansion of *A. agrarius* [20,69], in Lithuania the species presence increased after the 1990s, together with the expansion into commensal habitats [21]. There are no published data on the status of the species in Estonia, but the proportion of *A. agrarius* was 18% in 2021, 28% in 2022, 20% in 2022, 19% in 2023, and 22% in 2024. In Saaremaa, the proportion of *A. agrarius* was always high, sometimes up to 80% in cereal fields [22]. The above observations indicate that *A. agrarius*, a widely distributed species, is highly adaptable to climatic and environmental changes. As suggested by J. Cui et al., future studies should focus on widely distributed species and standard parameters, taking into account factors related to temperature [3].

Do observed changes in morphometry coincide with climates changes in analyzed countries? Using data from [70], we summarized average annual air temperatures for Slovakia, Lithuania, and Estonia from 1970 to 2023 (Appendix A). All three countries experienced significant warming, with Slovakia showing the highest increase (6.25 °C to 8.5 °C), followed by Lithuania (6.24 °C to 8.19 °C, peaking in 2020) and Estonia (5.13 °C to 6.8 °C, also peaking in 2020). This trend underscores notable regional warming over the past five decades, with 1991–2020 being consistently warmer than previous periods, as shown in Appendix A.

Annual precipitation trends from 1970 to 2023 show variability in Slovakia, Lithuania, and Estonia (Appendix A). Slovakia experienced a peak in 2020 followed by a slight decrease, while Lithuania showed variability with a gradual increase towards the end of the period. Estonia showed a steady increase over the decades. Appendix A highlights the long-term stability in Slovakia and Lithuania, with a notable upward trend in Estonia.

Lithuania and Estonia share a broadly similar climate due to their Baltic Sea location and geographical proximity. Slovakia, while also in a continental climate zone, has more temperature extremes, hotter summers, and greater climatic variation due to its inland and mountainous characteristics.

Climate change scenarios for Central Europe, including Slovakia, predict a 2–4 °C rise in 30-year average temperatures and a 10% increase in annual precipitation by 2100, with greater changes in the north [71,72]. Observations confirm rising annual temperatures, especially in February, April, and November, affecting bioclimatic indices and agriculture. In the Baltic Basin, significant warming has been observed, particularly from late spring to early autumn, alongside shifts in precipitation patterns, with more winter rainfall and reduced summer precipitation, especially in June and July [73]. Projections for the Baltic Sea region suggest a 3–5 °C temperature rise by 2100, depending on emissions, with the most pronounced increases in winter and greater precipitation in northern areas [74]. Earlier spring onset by 20–45 days aligns with March warming, and abrupt climatic events since the late 1980s have had cascading effects across ecosystems [75,76].

The morphometric changes we observed in *A. agrarius* closely follow regional climate trends, highlighting a link between environmental change and species adaptation. Consistent with Bergmann’s rule, the increase in body mass in northern populations corresponds to colder conditions historically observed at higher latitudes, while the reduction in appendage size (e.g., tails and ears) is consistent with Allen’s rule to minimize heat loss. Over time, rising temperatures in Slovakia, Lithuania, and Estonia, particularly during late spring, summer, and early fall [70], may have driven adaptations such as increased body and appendage sizes, reflecting responses to changing thermal and ecological pressures. In addition, shifts in precipitation patterns, including wetter winters and drier summers [70], have likely influenced habitat conditions and resource availability, further shaping the morphological traits of *A. agrarius*. To test James’ rule or the dependence of traits on climatic moisture [77], we would need additional data on changes in precipitation over decades in the sampling sites, which are currently not available. These coincidences suggest that climate change is an important driver of the observed geographic and temporal variation in the species’ morphology.

## 5. Conclusions

Our study highlights the interplay between climatic and geographic factors in shaping the size of *A. agrarius*. Results confirm significant morphological variation across geographic gradients: body size increases northward (Bergmann’s rule), while appendage size (tail, ears, and hind feet) decreases (Allen’s rule).Temporal changes in body size and appendages vary by region, with increases in Slovakia and mixed trends in Lithuania and Estonia. In Slovakia, however, local differences are also confirmed by previous studies.Sex-based dimorphism in *A. agrarius* varies by region and time period, with adult females generally heavier than males, while subadult males are often heavier. Dimorphism in appendage length is less pronounced.The species descriptions should be updated due to the changes in body and appendage size of *A. agrarius*, which have been particularly pronounced in the last few decades.

## Figures and Tables

**Figure 1 animals-15-00452-f001:**
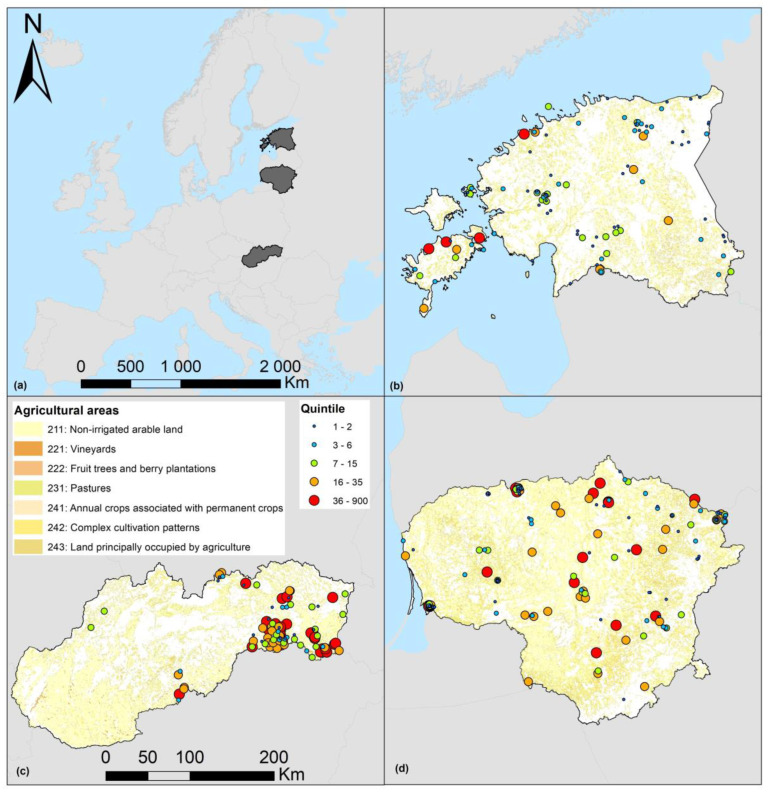
Position of the countries according to south–north and west–east gradients, shown in grey (**a**) and sampling sites in Estonia (**b**), Slovakia (**c**), and Lithuania (**d**). Trapping intensity is expressed as quintile with number of trapped *Apodemus agrarius* individuals. CORINE land use class 2 is shown as background (https://www.eea.europa.eu/data-and-maps/figures/corine-land-cover-1990-by-country/legend, accessed on 5 January 2025). System of coordinates: ETRS_1989_LAEA; Projection: Lambert_Azimuthal_Equal_Area; WKID: 3035 Authority: EPSG.

**Figure 2 animals-15-00452-f002:**
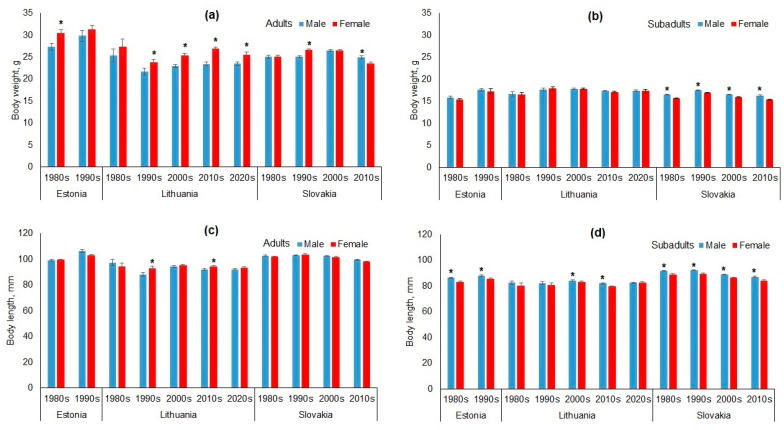
Sex-based size dimorphism in *Apodemus agrarius* over decades in Estonia, Lithuania, and Slovakia: body mass in adults (**a**) and subadults (**b**), and body length in adults (**c**) and subadults (**d**). Error bars indicate standard errors; asterisks indicate statistically significant differences between the sexes.

**Figure 3 animals-15-00452-f003:**
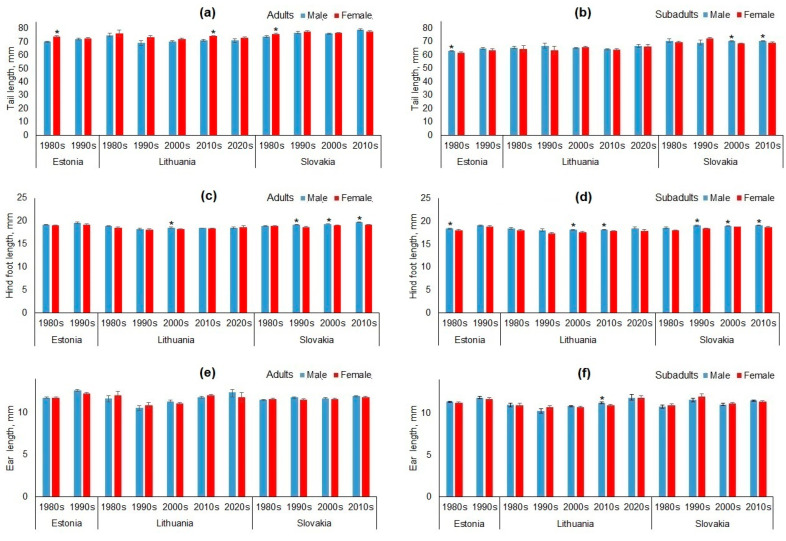
Sex-based appendage dimorphism in *Apodemus agrarius* over decades in Estonia, Lithuania, and Slovakia: tail length in adults (**a**) and subadults (**b**), hind foot length in adults (**c**) and subadults (**d**), and ear length in adults (**e**) and subadults (**f**). Error bars indicate standard errors; asterisks indicate statistically significant differences between the sexes.

**Figure 4 animals-15-00452-f004:**
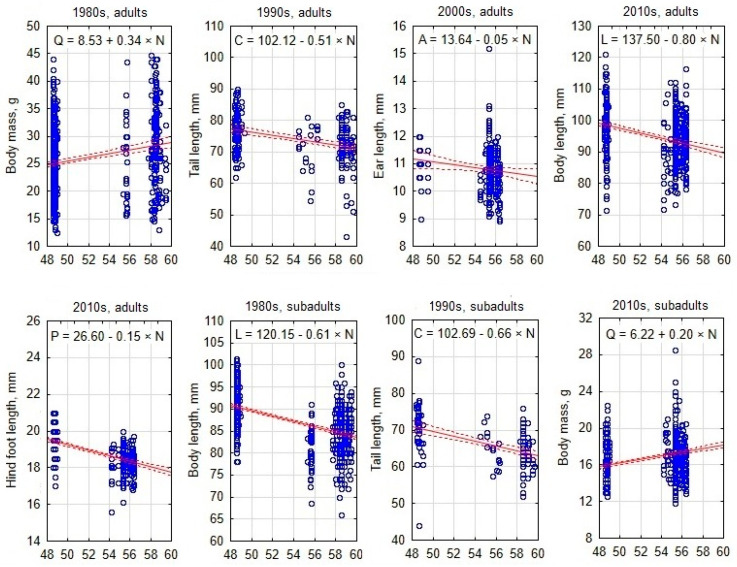
Examples of best-expressed south–north gradients in morphological trait values in adult and subadult *Apodemus agrarius* in different decades. Each blue dot represents an individual mouse, with trait values plotted against latitude, while the solid red lines indicate regression trends and dashed lines show 95% confidence intervals.

**Figure 5 animals-15-00452-f005:**
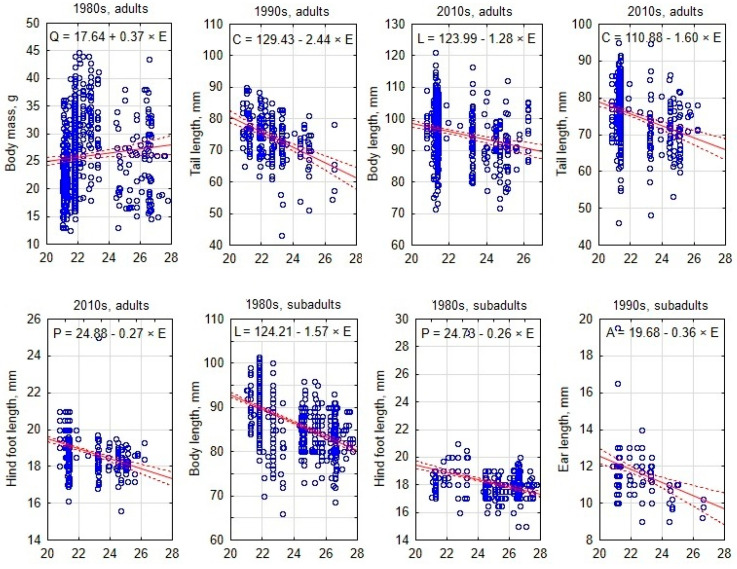
Examples of best-expressed west–east gradients in morphological trait values in adult and subadult *Apodemus agrarius* in different decades. Each blue dot represents an individual mouse, with trait values plotted against longitude, while the solid red lines indicate regression trends and dashed lines show 95% confidence intervals.

**Figure 6 animals-15-00452-f006:**
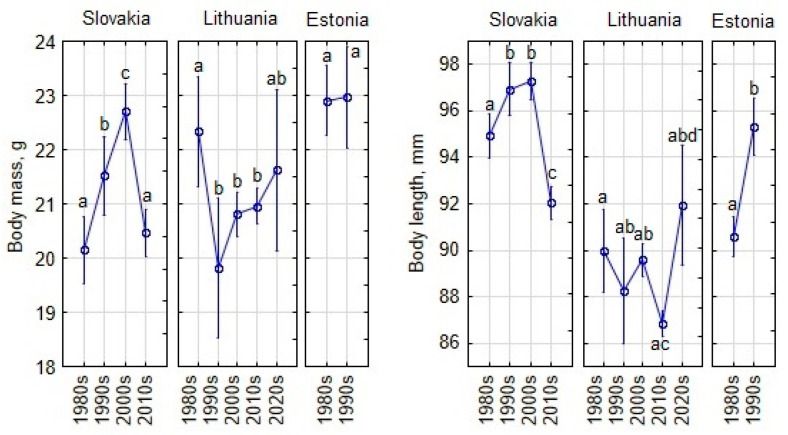
Temporal changes in body size of adult and subadult *Apodemus agrarius* by decade, calculated for covariates of sex and geographical position at their means. Vertical bars represent 0.95 confidence intervals. Statistically significant differences are indicated by different letters.

**Figure 7 animals-15-00452-f007:**
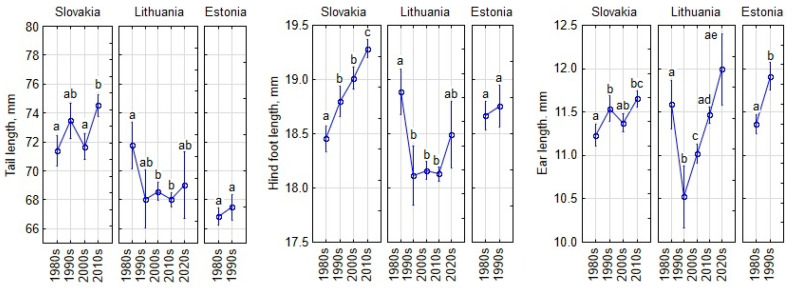
Temporal changes in appendage size of adult and subadult *Apodemus agrarius* by decade, computed for covariates of sex and geographic position at their means. Vertical bars indicate 0.95 confidence intervals. Statistically significant differences are indicated by different letters.

**Table 1 animals-15-00452-t001:** Characteristics of *A. agrarius* sample in Slovakia, Lithuania, and Estonia: N—sample size in total, M—males, F—females, ad—adult animals, sub—subadult animals, juv—juveniles, 1980s–2020s—decades.

Country	N	M	F	Ad	Sub	Juv	1980s	1990s	2000s	2010s	2020s
Slovakia	7004	3581	3291	2900	3392	319	1824	1677	2857	646	780
Lithuania	3823	2179	1595	788	1087	1906	95	274	879	1795	
Estonia	1101	613	460	415	443	154	836	265			

**Table 2 animals-15-00452-t002:** Distribution of normal and non-normal cases in males and females of *Apodemus agrarius* across morphological traits. Counts reflect the number of cases classified as normal or non-normal for each trait and sex, with age groups pooled.

Trait	Cases in Males		Cases in Females	
	Normal	Non-Normal	Normal	Non-Normal
Body mass, Q	30	68	43	55
Body length, L	58	40	55	43
Tail length, C	67	31	54	44
Hind foot length, P	45	52	39	59
Ear length, A	42	55	30	68

**Table 3 animals-15-00452-t003:** Univariate results (presented as F and *p*) of the factor influence on variation in morphological characteristics in *Apodemus agrarius*: Df—degrees of freedom, NS—not significant.

Factor	Df	Body Mass, Q	Body Length, L	Tail Length, C	Hind Foot Length, P	Ear Length, A
Latitude	1	F = 70.2, *p* < 0.001	F = 184.5, *p* < 0.001	F = 229.3, *p* < 0.001	F = 6.1, *p* < 0.05	F = 4.8, *p* < 0.05
Longitude	1	F = 35.4, *p* < 0.001	F = 0.1, NS	F = 3.8, *p* = 0.05	F = 205.9, *p* < 0.001	F = 25.1, *p* < 0.001
Decade	4	F = 16.8, *p* < 0.001	F = 97.2, *p* < 0.001	F = 3.8, *p* < 0.005	F = 24.8, *p* < 0.001	F = 28.3, *p* < 0.001
Age	2	F = 2374.1, *p* < 0.001	F = 2113.1, *p* < 0.001	F = 1115.4, *p* < 0.001	F = 300.3, *p* < 0.001	F = 305.7, *p* < 0.001
Sex	1	F = 1.30, NS	F = 29.1, *p* < 0.001	F = 1.3, NS	F = 74.3, *p* < 0.001	F = 4.7, *p* < 0.05

**Table 4 animals-15-00452-t004:** Correlations between morphological trait values and latitude in adult and subadult *Apodemus agrarius* by decade. NS—not significant.

Decade	Age	Body Mass, Q	Body Length, L	Tail Length, C	Hind Foot Length, P	Ear Length, A
1980s	ad	r = 0.21, *p* < 0.001	r = −0.20, *p* < 0.001	r = −0.21, *p* < 0.001	r = 0.08, NS	r = 0.10, *p* < 0.05
	sub	r = −0.08, *p* < 0.01	r = −0.55, *p* < 0.001	r = −0.40, *p* < 0.001	r = 0.02, NS	r = 0.17, *p* < 0.001
1990s	ad	r = 0.19, *p* < 0.001	r = −0.05, NS	r = −0.36, *p* < 0.001	r = 0.14, *p* < 0.05	r = 0.31, *p* < 0.001
	sub	r = 0.05, NS	r = −0.39, *p* < 0.001	r = −0.50, *p* < 0.001	r = 0.11, NS	r = −0.05, NS
2000s	ad	r = −0.17, *p* < 0.001	r = −0.34, *p* < 0.001	r = −0.23, *p* < 0.001	r = −0.33, *p* < 0.001	r = −0.22, *p* < 0.001
	sub	r = 0.28, *p* < 0.001	r = −0.33, *p* < 0.001	r = −0.37, *p* < 0.001	r = −0.44, *p* < 0.001	r = −0.14, *p* < 0.001
2010s	ad	r = 0.10, *p* < 0.05	r = −0.34, *p* < 0.001	r = −0.41, *p* < 0.001	r = −0.52, *p* < 0.001	r = 0.01, NS
	sub	r = 0.32, *p* < 0.001	r = −0.43, *p* < 0.001	r = −0.50, *p* < 0.001	r = −0.53, *p* < 0.001	r = −0.16, *p* < 0.001

**Table 5 animals-15-00452-t005:** Correlations between morphological trait values and longitude in adult and subadult *Apodemus agrarius* by decade. NS—not significant.

Decade	Age	Body Mass, Q	Body Length, L	Tail Length, C	Hind Foot Length, P	Ear Length, A
1980s	ad	r = 0.09, *p* < 0.02	r = −0.24, *p* < 0.001	r = −0.25, *p* < 0.001	r = −0.27, *p* < 0.001	r = 0.00, NS
	sub	r = −0.09, *p* < 0.005	r = −0.54, *p* < 0.001	r = −0.37, *p* < 0.001	r = −0.41, *p* < 0.001	r = −0.12, *p* < 0.02
1990s	ad	r = 0.05, NS	r = −0.19, *p* < 0.001	r = −0.43, *p* < 0.001	r = −0.17, *p* < 0.001	r = 0.05, NS
	sub	r = 0.08, *p* < 0.001	r = −0.37, *p* < 0.001	r = −0.43, *p* < 0.001	r = −0.20, *p* < 0.001	r = −0.34, *p* < 0.001
2000s	ad	r = 0.00, NS	r = −0.09, *p* < 0.005	r = −0.20, *p* < 0.001	r = −0.31, *p* < 0.001	r = −0.31, *p* < 0.001
	sub	r = 0.23, *p* < 0.001	r = −0.16, *p* < 0.001	r = −0.26, *p* < 0.001	r = −0.37, *p* < 0.001	r = −0.24, *p* < 0.001
2010s	ad	r = 0.03, NS	r = −0.23, *p* < 0.001	r = −0.30, *p* < 0.001	r = −0.37, *p* < 0.001	r = −0.03, NS
	sub	r = 0.07, NS	r = −0.19, *p* < 0.001	r = −0.21, *p* < 0.001	r = −0.34, *p* < 0.001	r = −0.09, *p* < 0.05

## Data Availability

The datasets presented in this article are not readily available because of different regulation of data ownership between individuals/institutions; legal reasons such as contracts; and because the data are part of an ongoing study.

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
