# Peer review of "Morphological Variation in the Striped Field Mouse Across Three Countries"

_animals, 2025, doi:10.3390/ani15030452_

Round 1

Reviewer 1 Report

Comments and Suggestions for Authors

Dear authors,

Congratulations on your work!

I believe some things could be improved, some clarifications regarding your work.

I missed a bit of discussion regarding your results and its implications. All is very superficial and I think you need more clear justifications. Your map needs major improvements, since it is not possible to distinguish land uses. Also, your statistics and results are quite messy and should be improved.

Best of luck

Author Response

Rev#1 comments and answers

Comment: I believe some things could be improved, some clarifications regarding your work. I missed a bit of discussion regarding your results and its implications. All is very superficial and I think you need more clear justifications. Your map needs major improvements, since it is not possible to distinguish land uses. Also, your statistics and results are quite messy and should be improved.

Answer: we understand that real comments are presented in the pdf attached, therefore answer these one by one, indicating line numbers.

Comment Line 31: Compared to your simple summary, this part is much less informative. Please add the same bit of information regarding these rules in here.

Answer: done as recommended

Comment Line 46: I don't know if I agree with this. Larger body size also allows species to explore landscape at larger scales, and so they have acess to more resources. Smaller species are much more vulnerable. You don't have a reference to support this claim, because usually is the other way around...

Answer: most possibly you have in mind larger species of animals. Larger body size of striped field mouse hardly will help much with landscape exploration possibilities. For the text you mention, we cited Kozłowski, J.; Konarzewski, M.; Czarnoleski, M. Coevolution of body size and metabolic rate in vertebrates: A life-history perspective. Biol. Rev. 2020, 95, 1393–1417. https://doi.org/10.1111/brv.12615.

Comment Line 55: The same as before, please give some more context about these rules.

Answer: text added.

Comment Line 63: Can you clarify this relation? It's not very clear what you mean

Answer: we added explanation from Hantak & Guralnik we cited, shown in capital letters here. “Urbanization effects have been shown to modify the influence of temperature on body size in mammals DUE TO BETTER RESOURCE AVAILABILITY [8], often resulting in larger individuals. “

Comment Line 149: Why is there a big jump in the last quantile? They should be more equilibrated

Answer: our apologies, these are quintiles. Quintiles divide a dataset into five equal parts, each containing 20% of the data. This is useful for understanding the distribution of data and identifying where certain values lie relative to others in a dataset. They are not equilibrated in the way you think; the last quintile also holds 20% of our data.

Comment Line 150: According to your collor scheme is nearly impossible to differentiate between land use categories. I strongly advise you to change the colors.

Answer: these colors are standard for CORINE: Corine land cover classes and RGB color codes — European Environment Agency, https://www.eea.europa.eu/data-and-maps/data/corine-land-cover-2/corine-land-cover-classes-and. It is RGB, and we cannot change the specification. On the other hand, maps are just illustrative, to show the distribution of agricultural areas across three countries. In fact, we should use only one color, to refer as “agricultural land”

Comment Line 161: Like I said before, it's not possible to see this in your map...

Answer: of course. It is said in the caption “CORINE land use class 2 is shown as background“. We show the most important habitat, however, no habitat analysis is done, so maps equally could show just trapping sites.

Comment Line 158: Change this, you're repeating the beggining of the previous sentence.

Answer: done

Comment Line 186: Sherman?

Answer: no, as it was said in line 185, snap traps were used, so the cannot be Sherman livetraps.

Comment Line 201: This belongs in the results section, and not metodology...

Answer: we would like to keep this as Sample size, and in Chapter 2. Our results are morphology.

Comment Line 212: The letters do not make sense to me, according to what you are measuring, so it's quite dificult to know which one is which for the rest of your text

Answer: even if you do not like these abbreviations, they are standard, used in hundreds of publications about small mammals, including ours. So, we would like to keep the compatibility.

Comment Line 218: Again, results

Answer: no, this text belongs to Material and methods. It is standard for papers about small mammals. Here we explain, how age groups were defined.

Comment Line 226: Results

Answer: as our results are morphometry and morphometric trends, we see habitat distribution as description of the sample.

Comment Line 253: These parametres have names, and you should use them before the respective letter...

Answer: we add “Fisher’s F-test” for F, but explaining p must be too much, it is standard.

Comment Line 262: Like I said before, this part of the text does not flow because I can't remember which one is which.

Answer: we were several times answering reviewer comments about NOT using abbreviations. Apologies, but these letters are common to scientists, working with small mammals

Comment in Table 2: Body mass – BM; Body lenght – BL; Tail lenght – TL; Hind foot lenght – HFL; Ear lenght - EL

Answer: The abbreviations Q (body mass), L (body length), C (tail length), P (hind foot length), and A (ear length) have been used in various morphological and zoological studies, particularly in mammalogy. However, their exact origin is not universally standardized across all scientific literature. These abbreviations were likely introduced by zoologists and morphometric researchers who needed shorthand notation for common body measurements in animals, especially rodents and small mammals. As we are using QLCPA in all our works, change is simply not possible.

Comment Figure 2: Is there any other way to plot this? The colors are not very appealing but the main issue I believe it is the fact that you have different coutries with different years in the same plot. Maybe chose different colors for the countries.

Answer: figure is devoted to show male-female differences. And, using blue for males and red for females is a common convention in many fields, including data visualization, biology, psychology, and social sciences. Of course, this color-coding can vary depending on context, culture, and discipline.

Used color scheme is shown in all parts of Figure 2 and Figure 3. Countries and decades are clearly shown on X axis. If we use different colors for every country, we (a) should triplicate male-female notation, and (b) – it will not help to see sex-based dimorphism.

If you really do not like blue and red used, we might try to use different kind of these colors, but please then give an example how these colors should look.

Comment Figure 4: I'm sorry to say but this figure is a total confusion. You should have all subadults in the same line, the same for adults, and all graphs should be organized by year. You have different measures, it's quite difficult to understand what are you trying to show. Please improve this

Answer: rearranged as requested – adults first, then subadults, both by decade.

Comment Figure 5: I understand not all graphs will best express the relation you want to show, but please try to organize them in a more obvious and clean way

Answer: rearranged as requested – adults first, then subadults, both by decade.

Comment Line 403: You are just stating your results again, and comparing to other studies. You have room for discussion, why those patterns happen?

Comment Line 432: Again, you are just stating your results and not exacly explaining them and give possible hypothesis for what is happening

Answer: as it is said in journal template, “Authors should discuss the results and how they can be interpreted from the perspective of previous studies and of the working hypotheses. The findings and their implications should be discussed in the broadest context possible. Future research directions may also be highlighted.

So we start from the ”state of art” in Lines 396-407, listing possible factors found by the other investigators. Further on, we present summaries of the results obtained, and compare these results with earlier studie in  respective countries, if any, and also in broader context.

In 4.3 we connect changes in morphological traits of Apodemus agrarius with changes of climate in respective countries. We, as the other researchers, are not sure about magnitude of climate change impact, therefore, using logical reasoning, in Lines 509-521, we relate changes in temperatures and precipitation to trends in morphometric traits. We use wording “likely” or ”may have driven”. As far as we know, there are no publications which “exactly explain” relation between climate change and morphology, results are not uniform.

Comment Line 411: Maybe this paragraph should be moved to the end, this is more about the dataset constrains and not about the discussion of your results

Answer: we used country order from south to niorth across the manuscript, so here we also start from Slovakia.

Comment Line 445: This paragraph is better, but could improve

Answer: thank you

Comment Line 471: In this particular case, I believe you can say the name of the author, otherwise it looks quite strange...

Answer: included, indeed much better

Comment Table 6: Did you use this in your model? Otherwise I don't understand its purpose here. Should be sup. information

Answer: no, we did not. Moved to Supplement as advised.

Reviewer 2 Report

Comments and Suggestions for Authors

On pages 4-5 there is a detailed description of the countries from which the mice were studied. This is largely unnecessary information, since it is not the countries that are important for mice, but the specific habitats. Please leave only the information that is directly necessary for characterizing the habitats.

In line 213, correct the letter designation for the ear.

In section 2.2. write how you accounted for pregnant females

It would be desirable for the authors to familiarize themselves with such works as

Panteleev P.A., 1994. Bergmann's rule - conceptual and empirical aspects // Uspekhi sovremen.biologii. Vol. 114. No. 1. P. 42–51.

Panteleev P. A., Terekhina A. N., Varshavsky A. A. & Bolshakov V. N. (1990). Ecogeographical variability of rodents. M: Nauka. 373 p.

Maxim Vinarski. 2013. On the application of Bergmann's rule to ectothermic organisms: The state of the art Zhurnal obshchei biologii 74(5):327–39

My main comment concerns the interpretation of the results obtained and their analysis.

The authors associate changes in body size and appendages with the temperature regime of the habitats of A. agrarius, which in their opinion corresponds to the Bergman and Allen rule. However, there is also a rule of hydrobiosis, which can also be the cause of changes in the size indicators of mice. Thus, populations of A. agrarius from Lithuania and Estonia live in a more humid climate than in Slovakia, which can lead to an increase in body size according to the rule of hydrobiosis. It is necessary to conduct a correlation analysis to determine the dependence of size indicators for temperature and separately for humidity.

Author Response

Rev#2 comments and answers

Comment: On pages 4-5 there is a detailed description of the countries from which the mice were studied. This is largely unnecessary information, since it is not the countries that are important for mice, but the specific habitats. Please leave only the information that is directly necessary for characterizing the habitats.

Answer: on the contrary, from our publishing experience we rather expected comment that countries should be characterized better. Coordinates are related to south-north and west-east gradients, we cannot avoid these. Th e rest – climate averages and habitat composition is presented by 1 paragraph per country. We use 300 words in total. There is nothing to shorten in this description.

Comment: In line 213, correct the letter designation for the ear.

Answer: apologies, corrected.

Comment: In section 2.2. write how you accounted for pregnant females

Answer: In pregnant females, the weight of the uterus and embryos was excluded from the body mass measurements. We added explanation to the text.

Comment: It would be desirable for the authors to familiarize themselves with such works as

Panteleev P.A., 1994. Bergmann's rule - conceptual and empirical aspects // Uspekhi sovremen.biologii. Vol. 114. No. 1. P. 42–51.

Panteleev P. A., Terekhina A. N., Varshavsky A. A. & Bolshakov V. N. (1990). Ecogeographical variability of rodents. M: Nauka. 373 p.

Maxim Vinarski. 2013. On the application of Bergmann's rule to ectothermic organisms: The state of the art Zhurnal obshchei biologii 74(5):327–39

Answer: our apologies, we checked the first and the third source, but failed to find what is special in them with relation to morphometry of A. agrarius. References [3] to [7] that we cited, are much more influential and have relation to the theme of manuscript. The book you would like us to cite is in Russian, and not available to us.

Comment: My main comment concerns the interpretation of the results obtained and their analysis. The authors associate changes in body size and appendages with the temperature regime of the habitats of A. agrarius, which in their opinion corresponds to the Bergman and Allen rule. However, there is also a rule of hydrobiosis, which can also be the cause of changes in the size indicators of mice. Thus, populations of A. agrarius from Lithuania and Estonia live in a more humid climate than in Slovakia, which can lead to an increase in body size according to the rule of hydrobiosis. It is necessary to conduct a correlation analysis to determine the dependence of size indicators for temperature and separately for humidity.

Answer: apologies, but we cannot accept your point of view. Under current understanding, hydrobiosis is the ability of an organism to survive extreme dehydration and then resume normal activity when water becomes available again. It is manifested in certain organisms like tardigrades, rotifers, and nematodes, as well as in some plant seeds and spores. Hydrobiosis is not found in small mammals or any vertebrates, as they lack the biological mechanisms necessary for true hydrobiosis. In mammals, severe dehydration can cause irreversible organ damage. We of course know, that mammals, like jerboas and kangaroo rats, can survive without drinking water, extracting moisture from food and minimizing water loss through highly efficient kidneys or minimize water loss through nocturnal lifestyles and specialized metabolic processes. However, in the countries covered by investigation, there are no such species, therefore, we cannot acknowledge your comment on hydrobiosis.

Round 2

Reviewer 2 Report

Comments and Suggestions for Authors

There is a mistake in the text of my comment. Naturally, it is not hydrobiosis that is meant, but hydrobiontic. The hydrobiontic rule is one of the ecogeographical rules according to which mammals belonging to the same taxon have, on average, larger body sizes in a humid climate than in a dryer one. In my opinion, your answer shows that you are really not familiar with this rule, although you are trying to analyze the patterns in which this rule can manifest itself. In addition, I asked for a correlation analysis to be carried out for specific environmental factors (temperature and humidity). To which you did not even deign to respond. You also brushed aside the literature, which analyzes some theoretical aspects of the application of the Bergman, Allen, and hydrobiontic rules.

Ignorance of the language is not a reason to ignore the relevant literature.

Author Response

Reviewer#2 comment Round 2

Comment: There is a mistake in the text of my comment. Naturally, it is not hydrobiosis that is meant, but hydrobiontic. The hydrobiontic rule is one of the ecogeographical rules according to which mammals belonging to the same taxon have, on average, larger body sizes in a humid climate than in a dryer one. In my opinion, your answer shows that you are really not familiar with this rule, although you are trying to analyze the patterns in which this rule can manifest itself. In addition, I asked for a correlation analysis to be carried out for specific environmental factors (temperature and humidity). To which you did not even deign to respond. You also brushed aside the literature, which analyzes some theoretical aspects of the application of the Bergman, Allen, and hydrobiontic rules.

Answer: sure, there was a mistake about hydrobiosis.

But there is also mistake in your current comment. The Hydrobiontic Rule refers to an ecological principle that describes the relationship between an organism’s dependence on water and its structural and physiological adaptations. It is primarily used in botany and plant ecology. Attribution of the Hydrobiontic Rule is not valid for mammals.

If you are specifically referring to humidity rather than temperature, the more relevant rule would be James’ Rule, which suggests that mammals in humid climates tend to be larger than their counterparts in drier climates. This is likely due to increased resource availability and thermoregulatory adaptations.

We however, did not find any references of the rule related to mice. The one citing original source (James, 1970) is that by Henry, E., Santini, L., Huijbregts, M. A., & Benítez‐López, A. (2023). Unveiling the environmental drivers of intraspecific body size variation in terrestrial vertebrates. Global Ecology and Biogeography, 32(2), 267-280. https://doi.org/10.1111/geb.13621, but it also is not directly related to our manuscript.

However, to acknowledge your comment, it is cited in the end of Discussion.

About correlation analysis you would like to be added. We see it out of the scope of the paper, as we analyze differences on the country level, and do not cover full geographic range of the species. In each country, there are at least some differences in temperatures and precipitation, data on which are not available on the site level for all decades we cover. We will gladly reply to your comment in the future, if it will be possible to include more different countries with A. agrarius presence, and if climatic data matrix will cover site level in different countries and decades.

Comment: Ignorance of the language is not a reason to ignore the relevant literature.

Answer: first we repeat our former answer, which was “we checked the first and the third source, but failed to find what is special in them with relation to morphometry of A. agrarius. References [3] to [7] that we cited, are much more influential and have relation to the theme of manuscript. The book you would like us to cite is in Russian, and not available to us.”

You recommended reference: Panteleev P. A., Terekhina A. N., Varshavsky A. A. & Bolshakov V. N. (1990). Ecogeographical variability of rodents. M: Nauka. 373 p.

In the answer, there was nothing- yes, NOTHING – about the ignorance of the language. Just as we know, this book is in Russian. That’s all about language.

We, however, also indicate, that already cited references are more related to the manuscript. In connection with this, we also would like to refer to Reviewer Guidelines:

  • Please ensure your comments are detailed so that the authors may correctly understand and address the points you raise.
  • Reviewers must not recommend citation of work by themselves, close colleagues, another author, or the journal when it is not clearly necessary to improve the quality of the manuscript under review.
  • Reviewers must not recommend excessive citation of their work (self-citations), another author’s work (honorary citations) or articles from the journal where the manuscript was submitted as a means of increasing the citations of the reviewer/authors/journal. You can provide references as needed, but they must clearly improve the quality of the manuscript under review.

So far, we did not understand why these references are so hard recommended, as none of the reasons were presented.